# Crystallization of molecular layers produced under confinement onto a surface

Jincheng Tong [1] ✉, Nathan de Bruyn[1], Adriana Alieva[1], Elizabeth. J. Legge [2,3], Matthew Boyes[1], Xiuju Song[1], Alvin J. Walisinghe [4], Andrew J. Pollard [2], Michael W. Anderson[1,4], Thomas Vetter[5], Manuel Melle-Franco [6] & Cinzia Casiraghi [1] ✉

It is well known that molecules confined very close to a surface arrange into molecular layers. Because solid-liquid interfaces are ubiquitous in the chemical, biological and physical sciences, it is crucial to develop methods to easily access molecular layers and exploit their distinct properties by producing molecular layered crystals. Here we report a method based on crystallization in ultra-thin puddles enabled by gas blowing, which allows to produce molecular layered crystals with thickness down to the monolayer onto a surface, making them directly accessible for characterization and further processing. By selecting four molecules with different types of polymorphs, we observed exclusive crystallization of polymorphs with Van der Waals interlayer interactions, which have not been observed with traditional confinement methods. In conclusion, the gas blowing approach unveils the opportunity to perform materials chemistry under confinement onto a surface, enabling the formation of distinct crystals with selected polymorphism.

Liquids confined between solid boundaries organize in thin layers[1–3], called interfacial molecular layers (MLs). These interfacial layers play a crucial role as they are present in living systems and do control interface and surface chemistry, lubrication, crystallization and many nanoscale effects. Furthermore, they exhibit distinct properties as compared to the corresponding bulk liquid. Hence, a considerable amount of studies has been dedicated to the investigation of the properties of fluids at solid interfaces[4–12], where the MLs have been created by putting a liquid under molecular-scale confinement, typically by encapsulation within a nanoscale cavity[13–15], pore[9,16,17], pocket[18,19], channels[20–22], or between two surfaces[3,23,24]. Similar confinement-based methodologies have been widely used also to study crystallization of molecules from solutions[25–30] because the kinetics or thermodynamics of crystallization are known to change by restricting the dimensions of the system into one, two, or three directions[25].

In this work, we propose a different type of confinement approach with the aim to crystallize MLs, which is based on restriction of the molecules onto one surface[31], enabled by the use of gas blowing. This approach does not require to confine a liquid between two surfaces[32], which would make difficult to directly access the crystals, or time-consuming and challenging fabrication of delicate nano-capillaries devices[21,22]. Traditionally, in crystallization studies the formation of crystals on a planar substrate is not considered to be confined[25]. However, here we use the gas blowing to force molecules to crystallize inside ultra-thin puddles, effectively restricting the volume of crystallization into two-dimensions. This gives rise to crystals that have not been observed with traditional confinement approaches. In particular, we demonstrate individual molecular layered crystals with well controlled structure, shape and thickness (from few-layers down to single-layer), which can be characterized with simple techniques, such as

[1]Department of Chemistry, University of Manchester, Manchester M13 9PL, UK. [2]National Physical Laboratory, Teddington, Middlesex TW11 0LW, UK. [3]Advanced Technology Institute, University of Surrey, Guildford, Surrey GU2 7XH, UK. [4]Curtin Institute for Computation, School for Molecular and Life Sciences, Curtin University, Perth, WA 6845, Australia. [5]Department of Chemical Engineering and Analytical Sciences, University of Manchester, Manchester M1 3AL, UK. [6]CICECO—Aveiro Institute of Materials, Department of Chemistry, University of Aveiro, Aveiro 3810-193, Portugal. ✉e-mail: tongjincheng@outlook.com; cinzia.casiraghi@manchester.ac.uk

Raman spectroscopy, and used for further processing into devices. Our approach enables the production of molecular layered crystals from crystallization of small molecules that exhibit layered polymorphs and can be used for the production of nanocrystals with controlled size and structure from various solution-processed nanomaterials.

## Results and discussion

As a proof of concept, we first performed crystallization under confinement onto a surface by gas blowing of glycine aqueous solutions. This molecule has been selected because of its simple structure and for its well-known polymorphs[33,34]: α-, β- and γ-form, with the latter being the most stable. The β-form is metastable at ambient conditions, but the α-form, which typically shows a pyramidal shape, is kinetically favored in aqueous solution. The β- and γ-forms are characterized by a very high piezoelectric coefficient[35,36], making these crystals attractive also for practical applications. Spatial confinement has been widely

applied to glycine molecules[27,28,37,38]. In particular, solution shearing of glycine demonstrated a clear transition between the α- and β-form, with the β-form being stabilized under 2-Dimensional (2D) confinement (i.e. at highest speed), giving rise to films of thickness of ~15 nm[39].

Figure 1a shows a schematic of the gas blowing confinement setup[40]: a gas knife is used to supply a uniform laminar gas-flow after the drop casting of the solution. Figure 1b shows the crystals obtained using a concentration of 0.01 M glycine aqueous solution by changing the pressure of the gas flow. At the pressure of 0.5 bar, thin films composed of aggregates of nanoparticles are observed using Atomic Force Microscopy (AFM). At the pressure of 1 bar, isolated nanoparticles with clear facets appear. Note that this morphology is very different from that obtained from simple drop-casting (Supplementary Fig. 4), where thick dendritic crystals are formed, indicating that the shearing generated by gas blowing plays an important role in determining the shape of the crystals, as already observed in our previous work on organic semiconductors[40].

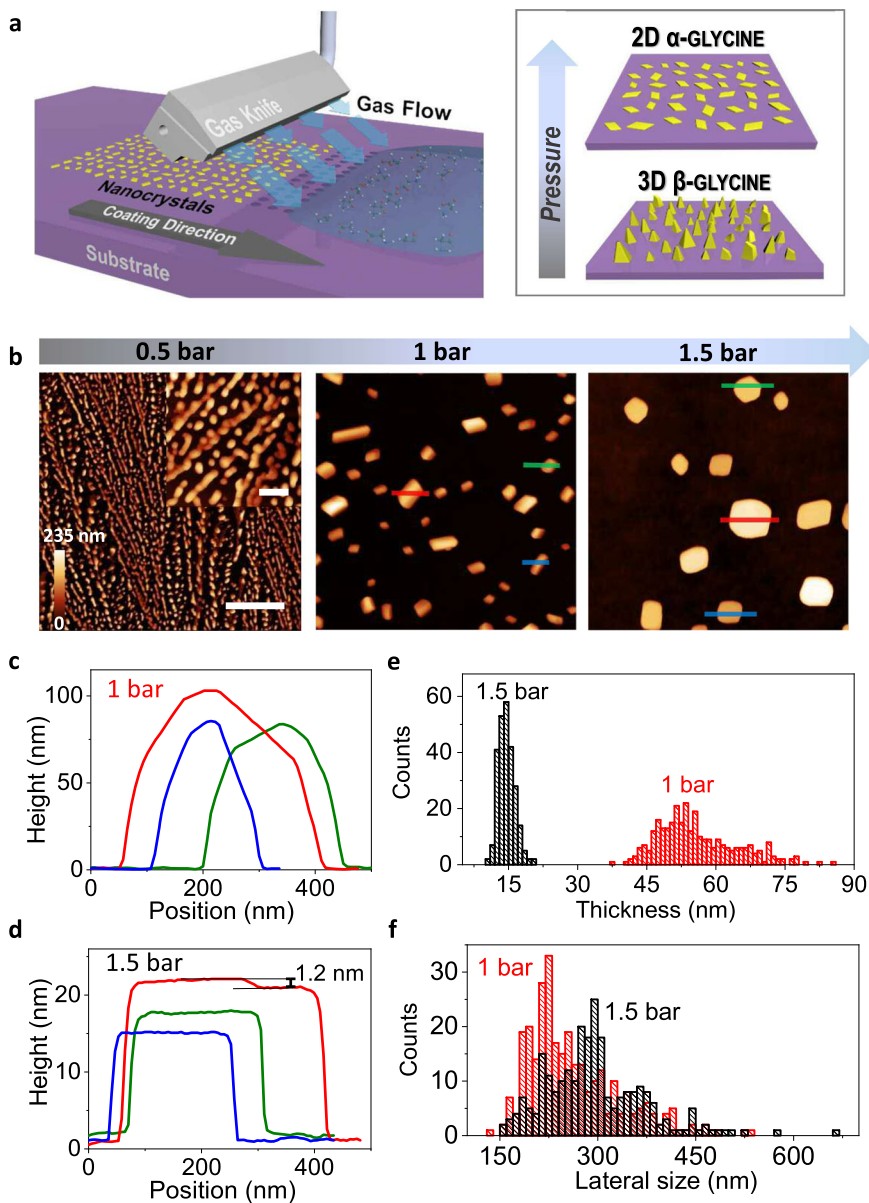

**Fig. 1 | Gas blowing enables controlled crystallization of glycine under confinement onto a surface. a** Schematic of the experimental setup. Gas pressure and solution concentration can be used to tune the crystal morphology and polymorph outcome. **b** Images taken by AFM of the crystals deposited for increasing gas pressure (from left to right). Scale bar = 20 μm; Inset scale bar = 2 μm. Cross section profiles of three selected crystals from panel **b** (indicated by the blue, red and green lines), grown at 1 bar (**c**) and 1.5 bar (**d**), respectively, and related thickness (**e**) and lateral size (**f**) distributions obtained by measuring more than 200 crystals.

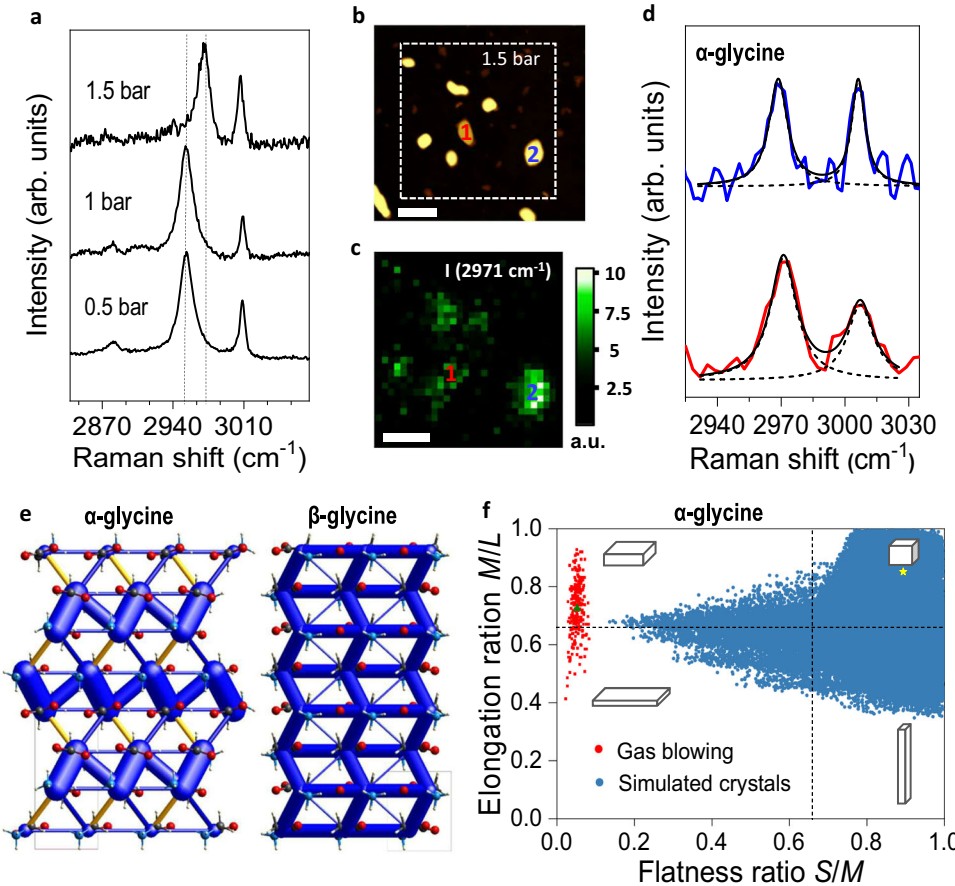

**Fig. 2 | Characterization and modeling of glycine crystals deposited by gas blowing at different pressures. a** Raman spectra of the crystals deposited by gas blowing of 0.01 M aqueous solution at different pressures. **b** AFM image of the glycine nanosheets (deposited at 1.5 bar); scale bar: 500 nm. **c** Raman map of the intensity (taken as height) of the peak at 2971 cm⁻¹, taken in the dotted square shown in panel **b**. Pixel size: 67 nm, integration time: 401 s, scale bar: 500 nm. **d** representative Raman spectra of the individual crystals 1 (in red) and 2 (in blue), shown in panels **b** and **c**. **e** Energy frameworks highlighting the stronger inter-molecular interactions for glycine crystals. α-glycine (left) 3 × 2 × 3 supercell showing three bilayer planes held by 2D contact interactions and β-glycine (right) 3 × 3 × 4 supercell showing the 3D network of strong contact interactions. Blue-colored cylinders represent binding interactions, while yellow cylinders highlight repulsive interactions, in both cases the radii of the cylinder are proportional to the strength of the interaction, only interactions larger in absolute value than 15 kJ/mol are shown. **f** Zingg Diagram of all the possible morphologies of α-glycine simulated using CrystalGrower (blue points, details in Supplementary Table 4). All crystals here are categorized using aspect ratio of Small:Medium (*S/M*) and Medium:Long (*M/L*) to define the shapes in the Zingg diagram. The schematic shows the four possible morphologies: plate (top left), lath (bottom left), block/sphere (top right), and needle (bottom right). The red points are showing the morphologies of the nanosheets obtained by gas blow coating at 1.5 bar (data in Supplementary Table 1). The green triangle marks the average size of the nanosheets. The yellow star corresponds to the morphology of a crystal obtained by slow evaporation of water at a supersaturation of 1.5 (Supplementary Fig. 13).

Statistical AFM analysis reveals that the nanoparticles have lateral sizes between 131 nm and 536 nm and thicknesses between 36 nm and 86 nm, giving rise to an average aspect ratio of 4.7 ± 0.8 (Fig. 1c, e, f; more details in Supplementary Fig. 5 and Supplementary Table 1). Thus, the shearing from the higher gas pressure has strongly affected the shape of the crystals, which are now quasi-2D. This can be understood by taking into account that by increasing the pressure, the initial droplet is likely to break into smaller droplets, hence the crystals have a higher probability to be formed through a single nucleation event from droplets that are sufficiently small, giving rise to isolated single crystals[41,42]. Further increasing the pressure to 1.5 bar leads to crystals with an atomically flat surface, sharp edges, a lateral size between 158 nm and 668 nm and a thickness between 10 nm and 21 nm, giving an average aspect ratio of 20 ± 5 (Fig. 1d, e, f; more details in Supplementary Fig. 5 and Supplementary Table 1). Two individual bilayers of glycine[42] with a terrace step of ~1.2 nm (Fig. 1d) can be clearly identified, indicating that the nanosheets were crystallized layer-by-layer. Therefore, our results show that an increase in the gas pressure causes the crystal dimensionality to change from quasi-2D to 2D.

Raman spectroscopy is used to study the polymorph outcome by measuring the characteristic C–H stretching frequencies[35,43]. The Raman spectrum of the α-glycine shows two peaks centered at 2972 cm⁻¹ and 3008 cm⁻¹, while the spectrum of the β-form shows two peaks at 2953 and 3009 cm⁻¹. Figure 2a compares the Raman spectra of the crystals shown in Fig. 1: the crystals obtained at pressure of 0.5 bar are β-glycine. The same polymorph is observed by increasing the pressure to 1 bar, in agreement with previous results obtained by solution shearing[39]. However, the crystals obtained at 1.5 bar are α-glycine, hence showing that the change in crystal morphology does correspond to a change in the molecular packing. Additional measurements were performed using AFM and confocal Raman mapping on the same crystals with a spatial resolution of ~200 nm[44]. Figure 2b shows the AFM image of the area investigated, while the corresponding Raman map is shown in Fig. 2c. Figure 2d shows two representative Raman spectra extracted from the Raman maps, indicated as 1 and 2 in Fig. 2b, showing that both crystals are α-glycine.

Our results show that both gas pressure and glycine concentration have strong effects on the shape and polymorphic form of the

crystals: in particular, a transition from β-glycine to α-glycine is observed by changing the pressure for a fixed concentration of 0.01 M. An increase in concentration leads to thicker and larger crystals of β-glycine, while a decrease in concentration leads to thin crystals of α-glycine with irregular shape (Supplementary Figs. 6, 7 and Supplementary Table 2). However, if the concentration is too low, only very small dots with thickness less than 2 nm and a lateral size well below 50 nm are formed, possibly associated to crystal nuclei (Supplementary Fig. 8). To note that the gas blowing technique has been previously used to produce large area (over mm) continuous thin films of organic semiconductors for electronics[40], achieved using low pressure and high concentrations. In contrast, our results show that crystallization in ultra-thin puddles onto a surface can be obtained only using very high pressure and low solution concentrations, so the deposition conditions need to be carefully optimized.

The gas blowing approach allows synthesis of α-glycine nanosheets crystals from solution. This result is in contrast with previous works reporting β-glycine to be mainly formed using traditional confinement approaches[27,28,37–39]. Furthermore, the β-glycine nanocrystals produced by gas blowing are stable under ambient conditions for at least 9 months (Supplementary Fig. 9). This result, together with the observation of individual bilayer terrace and flat surface shows that the α-glycine nanosheets produced at high pressure are not obtained as a result of the β-glycine nanocrystals turning into the α-form. Therefore, we attribute the formation of the α-glycine nanosheets to both confinement and shearing produced during the gas blowing crystallization: the strong shear produced by the high gas pressure causes the liquid film to spread and eventually break into thin puddles. The size of the contact between the droplet and the substrate, $l$, corresponds to the lateral size of the droplet and it is determined by the contact angle between the substrate and the liquid. However, the exact shape of the droplet results from a balance between the shear force (which favors a contact) and capillarity (which opposes it)[45]. Using the model developed for a droplet subjected to gravity[45], a droplet of initial radius R turns into a puddle when R is larger than the capillary length ($k^{-1}$), which is proportional to $\gamma/\rho \cdot F_s$, where $\gamma$, $\rho$ and $F_s$ are the surface tension, the density, and the shear force respectively. Thus, for increasing shear force (i.e. gas pressure), the droplet is more likely to turn into a puddle, thus enabling crystallization in a 2D space. This explains why nanosheets of glycine are seen at the high pressure of 1.5 bar. Note that nucleation under shear has been reported in the literature, typically to study crystallization in polymers, where the shear is continuously applied during crystallization[46,47]. However, in our case the effect of the shear is rather different as this is used to form ultra-thin puddles, allowing crystal nucleation to happen in each puddle, i.e. under 2D confinement.

To understand the origin of the selectivity of the α-crystals of glycine observed at high pressure, one has to remember that, although the β- and α-glycine crystals have similar structure, there is an important difference in their interlayer interactions. The α-glycine crystallizes in monoclinic space group $P2_1/n$, where the glycine molecules are in the form of centrosymmetric dimers. The dimers are linked with each other by hydrogen bonds along the a-c plane forming 2D hydrogen-bonded bilayers with a thickness of ~0.6 nm[48]. These bilayers stack together through the Van der Waals (VdW) force along the b direction. In contrast, β-glycine crystallizes in a chiral space group $P2_1$ and glycine monomers are linked with each other to form molecular layers along the a-c plane, while adjacent layers are linked together by hydrogen bonds along the b direction[38]. Both structures are layered, but α-glycine is a VdW crystal, while in the β-glycine the layers interact by H-bonding, which is stronger and anisotropic, in comparison with the VdW interactions. Therefore, under confinement in a 2D space, the α-glycine is likely to be energetically favorable because it is easier for a VdW layered crystal to accommodate shear in the direction parallel to the crystal planes and fit into a 2D space, as compared to an H-bonded crystal.

In order to confirm these observations, we analyzed the different energetics of α- and β-glycine crystals with the CrystalExplorer17 software[49]. First, we estimated the relative stabilization for crystals of increasing sizes (Supplementary Fig. 10). For the smaller nanocrystals, the β-form is predicted thermodynamically more stable, which might explain why β-glycine is typically found under 1D and 3D nanoconfinement conditions[27,28,37,38]. Interestingly, both polymorphs are built from a common one-molecule-thick monolayer with contact interactions ranging from −22 kJ/mol to −106 kJ/mol. This difference arises from the zwitterionic state of the glycine molecules, which shows highly anisotropic interactions. The relative orientation of the monolayers in the two polymorphs is different, yielding markedly different interactions. The strongest interaction, −186 kJ/mol, is found in α-glycine and is due to its centrosymmetric dimers. In fact, this large interaction holds two neighboring monolayers strongly together, which pile up, bound by one order of magnitude weaker dispersion forces. In contrast, the largest intermolecular interaction for β-glycine, −116 kJ/mol, connects the two neighboring layers resulting in a 3D network of strong contact interactions, Fig. 2e, thus explaining why the 2D confinement favors the crystallization of α-glycine in detriment of the β-form.

Figure 2f shows the Zingg Diagram of all the possible morphologies of α-glycine crystals obtained using CrystalGrower software. These simulations demonstrate that the morphological features of the α-glycine crystals obtained with our approach are completely different from the archetypal geometries typically obtained through bulk crystallization, irrespective of the level of supersaturation. It should be noted that these computational models were generated under the assumption of infinite volume, thus further supporting the hypothesis that the distinctive nanosheet morphology observed in our work are fingerprints of the confined crystallization conditions inherent to the gas blowing deposition method. Further calculations corroborate that the (010) facet appears to play a pivotal role in the emergence of these high aspect-ratio plate-like crystals. The growth rate analysis reveals that the [010] direction fails to surmount the requisite 2D nucleation barrier, thereby precluding alternative growth scenarios and substantiating its role in this distinct morphology (more information in Supplementary Section 3.2, Supplementary Figs. 11–14 and Supplementary Tables 3 and 4).

To further investigate the polymorph selectivity observed in glycine and to confirm our understanding of the fundamentals of the process, we applied crystallization by gas blowing to other molecules: benzamide, DL-methionine and D-mannitol. For each molecule, the deposition conditions were optimized following the same protocol used for glycine, as described in Supplementary Section 1 (Supplementary Figs. 1–4) and Supplementary Section 2. Similar to glycine, benzamide is characterized by two VdW layered polymorphs and one H-bond polymorph (Supplementary Section 4.1). Gas blow coating of 0.01 M benzamide solution at 1.5 bar gives rise to nanoplates with thickness of less than 50 nm (Supplementary Fig. 15). Raman spectroscopy indicates that the polymorph obtained is the stable VdW layered form-I. Furthermore, AFM measurements could identify a terrace step of ~1.2 nm, which can be assigned to the individual bilayer structure of form-I benzamide (Supplementary Fig. 16), in agreement with the results obtained with glycine.

DL-methionine was selected because both its α- and β-forms are layered VdW crystals with similar formation energy, but different stacking (Fig. 3a)[50]. AFM measurements show that single-layer nanosheets (1.5–2 nm in thickness) of DL-methionine can be obtained by gas blowing of 0.01 M DL-methionine solutions, Fig. 3b, in contrast to drop casting, which gives thicker crystals with different geometry, Supplementary Fig. 17. The layer-by-layer structure of DL-methionine can be clearly observed in Supplementary Fig. 18. Raman spectroscopy shows that the polymorph of the single-layer crystals is the metastable α-form, characterized by the thinnest

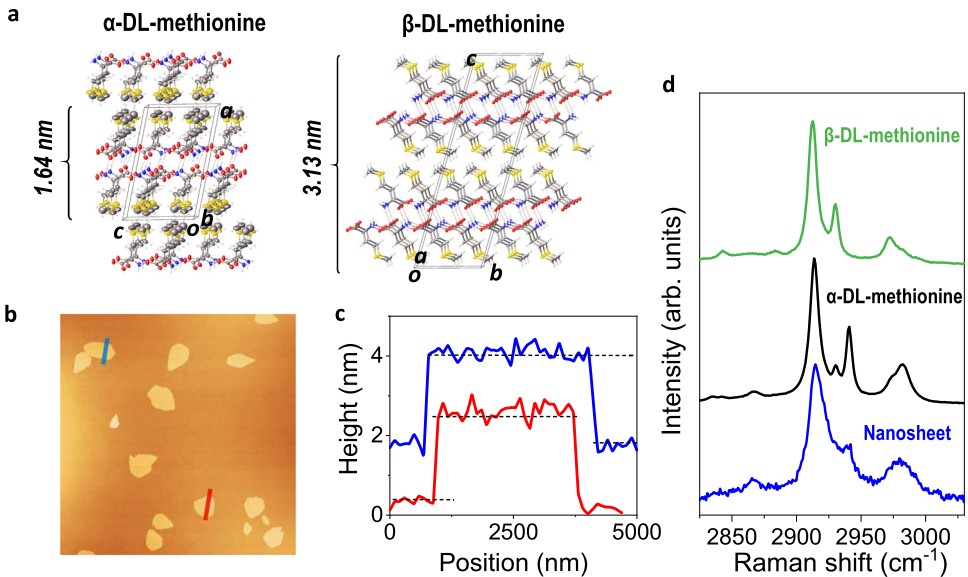

**Fig. 3 | Gas blowing of DL-methionine. a** 3D molecular packing showing the α- and β-form of DL-methionine. C: gray, H: white, N: blue, O: red, S: gold; intermolecular hydrogen bonds are depicted by red dashed lines. **b** Images taken by AFM of the nanosheets deposited by gas blowing of 0.01 M aqueous solution at 1 bar and (**c**) corresponding height profiles of two crystals, indicated by the red and blue lines in panel **b**; the blue cross section line is shifted upwards by ~2 nm for clarity. **d** Raman spectrum of an individual nanosheet, showing that the crystals are α-DL-methionine, as compared to the Raman spectra of the α-form (in black) and β-form (in green) of DL-methionine crystals obtained by drop casting.

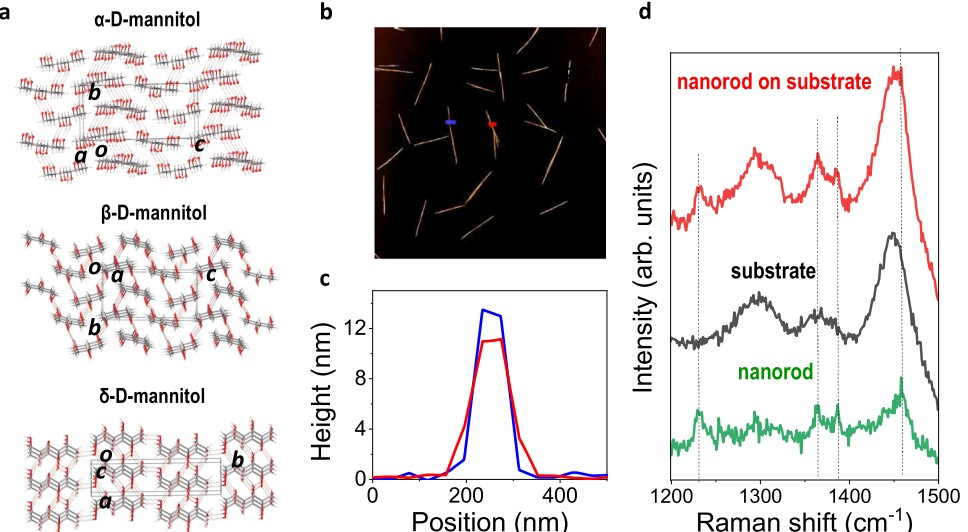

**Fig. 4 | Gas blowing of D-mannitol. a** 3D molecular packing showing the α-, β- and δ-form of D-Mannitol. C: gray, H: white, O: red and intermolecular hydrogen bonds are depicted by dashed lines. **b** AFM image and (**c**) corresponding height profiles of two selected crystals deposited by gas blow coating of 0.0025 M aqueous solution at a pressure of 1 bar, indicated by the blue and red lines in panel **b**. **d** The Raman spectrum of an individual nanorod on silicon. The Raman spectrum of the silicon substrate (in black) was also measured for reference and was subtracted from the overall spectrum (in red) for clarity. The Raman spectrum in green shows the typical peaks associated to the β-form of D-mannitol.

monomer (Fig. 3c, more details in Supplementary Section 4.2 and Supplementary Fig. 19).

Finally, D-mannitol is selected because it can form only H-bonded monomers (Fig. 4a). In this case, gas blowing of D-mannitol leads to the production of ultra-thin nanorods, instead of nanosheets, Fig. 4b. The smallest thickness obtained is ~10 nm, while the nanorods' width is ~100 nm (Fig. 4c). Raman spectroscopy shows the crystals to be β-form D-mannitol, in contrast to drop casting that gives rise to mixed polymorphs, Supplementary Fig. 20. Note that the Raman signal is relatively weak, but removal of the silicon background allows visualization of the Raman peaks of the β-form of D-mannitol (Fig. 4d). The 1D

morphology obtained by gas blowing is the result of the strong anisotropy of the H-bonding between the monomers of the β-form (Fig. 4a, d), which leads the crystal to grow preferentially in one direction. This shows that the gas blowing technique can also be used to grow H-bonded layered crystals with selected polymorphs, if no VdW polymorph exist, but the crystals shape will be ultimately determined by the directionality of the H-bonds.

In summary, we have shown a low-cost technique based on confinement onto a surface enabled by gas blowing that is able to produce molecular layered crystals. Our approach paves the way for a better understanding of the properties of molecular thin layers, and can be

potentially extended to other molecules (Supplementary Section 4.4 and Supplementary Fig. 21 for results on the crystallization of MOF-5 nanocrystals with thickness below 30 nm), hence providing an effective way to perform materials chemistry under confinement onto a surface.

## Methods

### Materials

Glycine (reagent plus, ≥99% purity), Benzamide (≥99% purity), DL-methionine (≥99% purity), D-mannitol (≥98% purity), Zinc nitrate hexahydrate (≥98% purity), 1,4 benzenedicarboxylic acid (≥98% purity), Acetone (99.5% purity), Isopropanol (anhydrous, 99.5% purity), 1-Butanol (anhydrous, 99.8% purity) and N,N-Dimethylformamide (anhydrous, 99.8% purity) were purchased from Sigma-Aldrich and used as received. Water (HPLC) was purchased from Fisher Scientific. The $SiO_2$/Si substrates and the glass substrates were bought from IDB Technologies Ltd and were cleaned by bath sonication in acetone and isopropanol for 5 min, respectively, and then dried with nitrogen. Argon plasma treatment was applied before deposition by gas-blow coating. A low pressure plasma system (Pico from Diener Electronic, Germany) was used with the pressure of 0.1 mbar at the power of 100 V for 2 min.

### Deposition

The gas blowing coating system is based on a doctor blade coater (Kpaint applicator purchased from RK Printcoat Instruments), equipped with a gas knife (purchased from Exair). A $N_2$ cylinder is used to supply the gas, whose pressure is controlled by a regulator. A breather check valve is applied to control the start or finish of the gas-blow coating process. The speed can be controlled by the speed control panel. The distance (5 mm) and the angle (57°) of the gas knife to the substrate were kept constant. The substrate was placed about 5 cm away from the front of the gas knife. An aqueous solution (2 µL) of glycine or D-mannitol was drop casted on the substrate and held for 5 s to allow the droplets to spread on the substrate before turning on the gas knife. For DL-methionine, 0.2 µL aqueous solution was used. For benzamide, 1 µL solution in 1-butanol was used. For MOF-5, 1 µL solution in N,N-Dimethylformamide was used. The instrument and the gas knife were turned on at the same time with the moving speed set at 6 mm/s. A film was deposited with the laminar gas passing over the substrate. The samples were then stored in a vacuum chamber at room temperature overnight before characterization.

### Characterization

A Nikon Eclipse LV100 microscope with different objectives was used to take optical images. Atomic force microscopy was conducted using a Bruker Nanoscope V with a Multimode 8 in tapping mode. The cantilevers were silicon tips on nitride lever with a nominal spring constant of $0.4 \, N \, m^{-1}$. The analysis of the AFM images was performed by using the Gwyddion software. The lateral size ($L$) and the thickness ($h$) distributions were obtained by taking the maximum bounding size and the average thickness value from individual crystals, respectively. The aspect ratio was obtained as: $L/h$. Raman spectroscopy was conducted with a Renishaw inVia Raman spectrometer equipped with a laser operating at 514.5 nm. The measurements were performed with a 100× objective, 2400 l/mm grating and the laser power was well below 1.5 mW. Streamline™ maps were taken with the step size of 0.4 µm. Co-located confocal AFM and Raman maps were obtained by using a transmission-mode TERS system consisting of an AFM (Combiscope, AIST-NT, USA) secured on the top of an inverted confocal optical microscope (Ti-U Eclipse, Nikon, Japan) coupled to a Raman spectrometer (iHR 320, HORIBA Scientific, France) and a charge-coupled device detector (Newton, Andor, Ireland). A 532 nm excitation laser (~450 µW power at the sample) was focused on the sample using a 1.49 NA, 100× oil immersion microscope objective (Apo TIRF, Nikon,

Japan), with a liquid crystal radial polarizer (Arcoptix, Switzerland) placed in the optical path to convert the linearly polarized laser beam to a radially polarized beam. The co-located AFM measurements were performed in a tapping-mode configuration (Nanosensors PPP-NCHR probe, NanoWorld AG Switzerland) with a set point of 85% and scan rate of 0.1 Hz.

### Intermolecular energies of glycine polymorphs calculations

The intermolecular energies of glycine were computed with Crystal-Explorer17 (CE) with the model CE-B3LYP based on scaled B3LYP-6-31g(d,p) calculations. Calculations were performed on structures retrieved from the Cambridge Crystallographic Data Centre (CCDC) database for polymorphs obtained under similar conditions (α: CCDC 1416370 refcode: DOLBIR07 and β: CCDC 1416371 refcode: DOLBIR08). Energies between molecular pairs were represented with CE's energy frameworks which use cylinders joining the centers of mass of the molecules with radius proportional to the magnitude of the interaction energy[51]. The frameworks were rendered with a scale factor of 50 and a cutoff value of 15 KJ/mol.

### Glycine morphology mapping by CrystalGrower

CrystalGrower was used to predict all possible morphologies of α-glycine (CCDC 1169354 refcode: GLYCIN02). The number of nearest neighbor interactions (NNI) was cut down to 11 interactions in total, and then coupled into 7 different NNI's that share both interaction type, surface area and symmetry (further details of the NNI's can be found in the Supplementary Information Section 3.2). Each NNI energy was varied from 1.0 to 3.0 kcal/mol at a step size of 0.5 kcal/mol, combinatorically, creating a series of 78,125 simulations. Crystals were simulated for 5 million iterations: 4 million iterations at high supersaturation, an equilibration period of 100,000 iterations and grown at equilibrium for the remaining iterations. Further details about using CrystalGrower can be found in references[52,53].

### Crystal structures of benzamide, DL-methionine, D-mannitol and MOF-5

The crystal structures were retrieved and adapted from CCDC database: form-I benzamide (CCDC 1118065 refcode: BZAMID), form-II benzamide (CCDC 267634 refcode: BZAMID06)); α-DL-methionine (CCDC 1208063 refcode: DLMETA07), β-DL-methionine (CCDC 270574 refcode: DLMETA05); α-D-mannitol (CCDC 224658 refcode: DMANTL08), β-D-mannitol (CCDC 224659 refcode: DMANTL09), δ-D-mannitol (CCDC 224660 refcode: DMANTL10) and MOF-5 (CCDC 2229855 refcode: ICSD 144277).

## Data availability

All data supporting the findings of this study are available within the paper and its Supplementary Information. All raw data for the current study are available from the corresponding authors upon request.

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

## Acknowledgements

J.T. thanks Michael L. Turner, Amadou Doumbia and Jingzhen Du for useful discussion. A.J.P. acknowledges Keith Paton for discussions on the

manuscript. C.C. acknowledges useful discussions with Kostya Novoselov, Paolo Samori and Laura Fumagalli on the manuscript. This work is supported by the European Research Council (ERC) under the European Union's Horizon 2020 research and innovation programme under grants agreement No 648417 and by the UKRI (Grant EP/X028844/1). J.T. acknowledges the University of Manchester for the President Scholarship Award (PDSA). N.d.B acknowledges support from the ESPRC (i-Case Award). E.J.L and A.J.P. acknowledge funding from the National Measurement System of the Department of Business, Energy and Industrial Strategy (BEIS), U.K, from Grant 124089. E.J.L. acknowledges funding the EPSRC for funding a DOCCAT collaboration and support from Grants EP/R025304/1 and EP/L02263X/1. T.V. thanks the Royal Academy of Engineering for the support through an Engineering for Development research fellowship (Grant No. RF1516/15/22). M.M-F. acknowledges support from the project IF/00894/2015 and within the scope of the project CICECO-Aveiro Institute of Materials, UIDB/50011/2020, UIDP/50011/2020 & LA/P/0006/2020, financed by national funds through the FCT/MEC (PIDDAC) and from the European Union's Horizon 2020 Research and Innovation Programme under Grant Agreement No. 964593.

## Author contributions

The project was conceived and designed by C.C and J.T.; J.T. performed the depositions and performed Raman and AFM measurements, under the supervision of C.C. and T.V. Additional measurements were performed by A.A, X.S and M.B. M.M.F. performed the calculations on intermolecular energies of glycine. CrystalGrower calculations were performed by N.d.B. and A.J.W and sections on CrystalGrower were written by N.d.B under the supervision of M.W.A. E.J.L. performed the co-located confocal Raman-AFM measurements under the supervision of A.J.P. The manuscript has been written by J.T. and C.C. with contributions from all authors.

## Competing interests

The authors declare no competing interests.
