## [Peer Review File · Nature Communications]

Crystallization of Molecular Layers produced Under Confinement onto a SurfaceEditorial Note: This manuscript has been previously reviewed at another journal that is not operating a transparent peer review scheme. This document only contains reviewer comments and rebuttal letters for versions considered at *Nature Communications*. Mentions of prior referee reports and responses have been redacted.

REVIEWER COMMENTS

Reviewer #1 (Remarks to the Author):

In the revised manuscript, the author revised the introduction part of the manuscript so that it is clear that the main purpose of this study is not to present an alternative method for investigating crystallization under confinement with solid-liquid interfaces but to present a different approach.

In this respect, the author has responded to my previous comments appropriately and I think the manuscript is worth being published.

Reviewer #2 (Remarks to the Author):

I think the authors have done a wonderful job addressing the concerns raised by the referees, and I agree with the authors that the original Referee 3 seemed to misunderstand many of the details that were clearly laid out in the original submission. Overall, I believe this manuscript is an important and impactful piece of work that the supramolecular chemistry community will read with great interest. I recommend acceptance in its current form.

Reviewer #3 (Remarks to the Author):

The authors have modified the manuscript and provided responses to the referees, but there are few issues that still demand attention.

1. The authors should be aware that they need to write the manuscript for the community, not for themselves. Therefore, they should be aware that it should be written in a way that will convey their message to the community. For example, they had major criticism to my previous review, for statements I made like:

"Limited control of polymorphism is achieved only for glycine (controlled crystallization of alpha and beta polymorphs, based on the pressure of the gas blowing which effectively control the size of the puddles)"

And they corrected this saying that I missed the point, which is (according to their response):

"[REDACTED]"

"Selective crystallization" and "control of polymorphism" is the same in this context - crystallization of one polymorph. If there is some nuance that I have missed, then I am confident other people will miss it too. And, again, this will be read by a broad audience. If the authors are concerned that their main message is not being conveyed, they should revisit their writing.

2. I appreciate that the authors modified the title to show that the crystallization is not done "under gas confinement" in true sense, but it is simply a crystallization of organic molecules from thin layers. However, I am not convinced that "gas blowing" really deserves to be in the title at all. Is glass blowing the only method for making ML? If the same ML layers can be made by other methods, then the same results will be obtained without using any gas blowing. In that case, the gas blowing is irrelevant so why should it be in the title?

3. The authors praise this method as "simple" in the abstract, through the text, and in the conclusion. But they acknowledge in the response to the reviewers that it is challenging to optimize the crystallization case by case. I think the manuscript is misleading. They made it sound like it is a readily applicable, straightforward method. To be real, it does not matter if the method is simple or not IF the optimization is difficult. What is the worth of the method being simple if it requires tedious optimization of experimental conditions? The authors should be frank in the main manuscript and write about challenges that come with the optimization.

Response to the Reviewers' comments

Reviewer: #1

In the revised manuscript, the author revised the introduction part of the manuscript so that it is clear that the main purpose of this study is not to present an alternative method for investigating crystallization under confinement with solid-liquid interfaces but to present a different approach. In this respect, the author has responded to my previous comments appropriately and I think the manuscript is worth being published.

Answer: We would like to thank the referee for helping us to improve our work and to confirm that the manuscript has been changed accordingly and it is therefore suitable for publication.

Reviewer: #2

I think the authors have done a wonderful job addressing the concerns raised by the referees, and I agree with the authors that the original Referee 3 seemed to misunderstand many of the details that were clearly laid out in the original submission. Overall, I believe this manuscript is an important and impactful piece of work that the supramolecular chemistry community will read with great interest. I recommend acceptance in its current form.

Answer: We really would like to thank referee 2 for fully supporting the publication of our work and for appreciating the efforts we made in addressing the referees' comments and changing our manuscript accordingly.

Reviewer: #3

The authors have modified the manuscript and provided responses to the referees, but there are few issues that still demand attention.

Answer: we thank the referee for confirming that the text was modified according to the referees' comments, and for providing further suggestions.

1. The authors should be aware that they need to write the manuscript for the community, not for themselves. Therefore, they should be aware that it should be written in a way that will convey their message to the community. For example, they had major criticism to my previous review, for statements I made like:

Answer: we thank the referee for providing this advice. We agree indeed that writing of this manuscript is challenging because of the multi-disciplinary nature of the work, which ranges from supramolecular chemistry to crystallization from solution and 2D materials. We have tried our best to make the manuscript accessible to scientists coming from different communities, and we thank the referee for providing further suggestions.

"Limited control of polymorphism is achieved only for glycine (controlled crystallization of alpha and beta polymorphs, based on the pressure of the gas blowing which effectively control the size of the puddles)"

And they corrected this saying that I missed the point, which is (according to their response):

"[REDACTED]."

"Selective crystallization" and "control of polymorphism" is the same in this context - crystallization of one polymorph. If there is some nuance that I have missed, then I am confident other people will

miss it too. And, again, this will be read by a broad audience. If the authors are concerned that their main message is not being conveyed, they should revisit their writing.

Answer: we thank the referee for this observation, we think there was a misunderstanding maybe due to our answer not being well written. In our previous answer, we wanted to point out that we can select the glycine polymorph by using different deposition conditions, i.e. we do not grow a mixture of different polymorphs, so we have achieved polymorph control/selective crystallization.

The text has been modified as following:

Our results show that both gas pressure and glycine concentration have strong effects on the shape and polymorphic form of the crystals: in particular, a transition from β -glycine to α -glycine is observed by changing the pressure for a fixed concentration of 0.01 M.

2. I appreciate that the authors modified the title to show that the crystallization is not done "under gas confinement" in true sense, but it is simply a crystallization of organic molecules from thin layers. However, I am not convinced that "gas blowing" really deserves to be in the title at all. Is glass blowing the only method for making ML? If the same ML layers can be made by other methods, then the same results will be obtained without using any gas blowing. In that case, the gas blowing is irrelevant so why should it be in the title?

Answer: we thank the referee for this observation. As far as we know, nanosheets of α -glycine from solution crystallization have never been reported - there may be some other methods able to achieve the same nanosheets if similar confinement conditions can be reached without using the gas blowing, but this remains to be investigated and demonstrated. However, since the title is already quite long, we are happy to remove "gas blowing" from it.

3. The authors praise this method as "simple" in the abstract, through the text, and in the conclusion. But they acknowledge in the response to the reviewers that it is challenging to optimize the crystallization case by case. I think the manuscript is misleading. They made it sound like it is a readily applicable, straightforward method. To be real, it does not matter if the method is simple or not IF the optimization is difficult. What is the worth of the method being simple if it requires tedious optimization of experimental conditions? The authors should be frank in the main manuscript and write about challenges that come with the optimization.

Answer: our method is simple as compared to other techniques used to put molecules under confinement, such as nanocapillaries made of 2D materials, for example. This approach requires clean room access, expensive lithography and has a very low yield, i.e. most of the devices do fail. Indeed, very few groups are able to produce such samples, as demonstrated by a recent protocol published in Nature, reporting how to fabricate such nanocapillaries (Bhardwaj, A., Surmani Martins, M.V., You, Y. et al. Fabrication of angstrom-scale two-dimensional channels for mass transport. Nat Protoc (2023). <https://doi.org/10.1038/s41596-023-00911-x>). Other confinement techniques do not allow direct access to the crystals or are not suitable for fabrication of single and few layers of organic molecules. Our method requires to optimize two main parameters: gas pressure and concentration of the solution; if one changes the substrate (we used silicon), then wettability of the substrate should also be taken into account.

To clarify this point, we have changed the text as following:

Here we report a method based on crystallization in ultra-thin puddles enabled by gas blowing, which allows to produce molecular layered crystals with thickness down to the monolayer onto a surface, making them directly accessible for characterization and further processing.

For each molecule, the deposition conditions were optimized following the same protocol used for glycine, as described in Supplementary Section 1 and 2.

In summary, we have shown a low-cost technique, based on confinement onto a surface enabled by gas blowing, able to produce molecular layered crystals. Our approach paves the way for a better understanding of the properties of molecular thin layers, and can be potentially extended to other molecules (See Supplementary Section 4.4, for results on the crystallization of MOF-5 nanocrystals with thickness below 30 nm), hence providing an effective way to perform materials chemistry under confinement onto a surface.